# Inherent Bias in ROSA^®^ Zimmer Biomet Pre-Op Planning Using 2D to 3D X-Atlas^®^ Coronal Knee Axis Measurement

**DOI:** 10.3390/jcm14051698

**Published:** 2025-03-03

**Authors:** Michał A. Duchniewicz, Aly Shaaban, Manuel Müller, Philip M. Anderson, Lars Goebel, Patrick Orth, Milan A. Wolf, Felix Bachelier, Stefan Landgraeber, Philipp Winter

**Affiliations:** 1Department of Orthopaedic Surgery, University of Saarland, 66421 Homburg, Germany; manuelmueller@gmx.net (M.M.); lars.goebel@uks.eu (L.G.); patrick.orth@uks.eu (P.O.); stefan.landgraeber@uks.eu (S.L.); philipp.winter@uks.eu (P.W.); 2School of Clinical Medicine, University of Cambridge, Cambridge CB2 0SP, UK; 3Department of Orthopedics, Orthopädische Klinik König-Ludwig-Haus, University of Wuerzburg, 97074 Würzburg, Germany; 4Department of Orthopedics, Auguste-Viktoria-Klinik, Ruhr University Bochum, 44801 Bad Oeynhausen, Germany

**Keywords:** robotic knee replacement, total knee replacement, ROSA^®^ Zimmer Biomet, orthopaedic surgery, 2D to 3D X-Atlas^®^

## Abstract

**Background:** Robotic assistance is considered capable of improving precision and outcomes of total knee replacement. We assessed the inherent biases, pre-procedural planning accuracy using 2D to 3D X-Atlas^®^, and final knee axis outcomes of the ROSA^®^ Knee System (Zimmer Biomet, Warsaw, IN, USA). **Methods:** A total of 55 patients who underwent robotic-assisted knee replacement using ROSA^®^ Knee System (Zimmer Biomet, Warsaw, IN, USA) at a single center were included. Pre-procedural measurements performed by ROSA were compared to those performed by senior consultants. Component sizes predicted by ROSA^®^ were compared to those implanted. A final axis measurement was taken during the procedure. **Results:** Femur components were exactly matched in (83.64%) cases, accurately matched in a further 8 (14.55%), and inaccurately matched for only 1 (1.82%). Tibial component sizes were exactly matched by the planning for 39 (70.91%), accurately for 12 (21.82%), and inaccurately for 4 (7.27%). ANOVA did not show statistically significant differences between the predicted and implanted femur (*p* = 0.96) nor the tibia components (*p* = 0.27). We show that ROSA^®^ pre-procedural planning has a statistically significant bias (*p* = 0.001), with a deviation of 0.83 degrees into varus, when assessing the knee axis in the coronal plane, compared to senior consultant measurements. The average of the final coronal knee axis was 0.37 degrees in varus (SD = 2.49). **Conclusions:** ROSA^®^ accurately predicts implanted component sizes. Despite the small and statistically significant varus bias in initial knee axis assessment, the system results lay within the ±3° of neutral knee axis, which is the widely accepted knee replacement standard.

## 1. Introduction

Robotic surgical assisted technology (ROSA^®^) is a well-established method for total knee replacements (TKRs). Some studies suggest that this method has at least comparable outcomes to conventional TKR [1,2]. An analysis of 17 studies has shown that the ROSA system results in improved clinical outcomes within 1 year of surgery [3]. ROSA^®^-TKR has been shown to predict both femoral and tibial resections more accurately than the conventional methods [3]. This enhanced accuracy is comparable to that of other robotic-assisted technologies, such as the MAKO robotic arm-assisted technology, despite using two X-ray projections rather than a CT-scan for planning [4]. Coronal tibial and femoral cuts are predicted more accurately with ROSA^®^ compared to sagittal ones [1,5].

ROSA^®^-TKR was associated with shorter hospital stays, from 6.8 days in conventional TKR to 5.4 days, and better 6-month post-operative outcomes for pain and disability [1]. Moreover, the pre-operative 2D to 3D X-Atlas^®^ planning is included as part of ROSA^®^ Knee System, predicting the component size and describing the initial coronal knee axis. The system performs its measurements using whole-leg X-rays, with the patient wearing two strap attachments placed by a trained technician. The current literature does not report on the inherent biases of this system, particularly those related to pre-and post-TKR knee-axis alignment. The aims of the study were the following:To measure and quantify the accuracy of the pre-operative knee axis alignment measurement taken by the 2D to 3D X-Atlas^®^ planning compared to the current gold standard of whole-leg-length X-ray measurements performed by a senior consultant [6].To assess the accuracy of component size prediction compared to the implants used by the 2D to 3D X-Atlas^®^ pre-operative planning.

## 2. Materials and Methods

All patients in a single center who received the X-Atlas^®^ preoperative planning and underwent robotic-assisted primary total knee replacement between February 2022 and May 2024, using the ROSA^®^ Knee System (Zimmer Biomet, Warsaw, IN, USA), were included. In all cases, the Persona^®^ Knee System (Zimmer Biomet, Warsaw, IN, USA) cruciate retaining implant was used. Patients were included retrospectively based on the availability of the pre-operative X-Atlas^®^ (Zimmer Biomet, Warsaw, IN, USA) planning. Surgeries were performed by four high-volume (>100 arthroplasties per year) surgeons, following a medial parapatellar approach. Whole-leg X-ray images in both lateral and AP projections were taken prior to the operation. Optical guidance tools provided by Zimmer Biomet were attached to the patient’s leg by the X-ray technician. The images were taken at an appropriate resolution and with good exposure, allowing for the easy identification of bony landmarks and precise measurements. The same images used for 2D to 3D X-Atlas^®^ planning were used to perform knee axis measurement by a senior consultant, which is the current gold standard [6]. Measurements were taken from the center of the head of the femur, the anatomical center of the knee joint, and the anatomical center of the ankle joint. This was then compared to values obtained from X-Atlas^®^. Surgeons were blinded to this pre-operative X-Atlas^®^ axis measurement and component planning. This did not alter patient outcomes as the planning results were only produced after the planned procedure date, which was the standard practice at the center. The sizes of predicted and implanted components were recorded and compared. The accurate prediction of component sizes was defined as being within one size of the implanted component size, while exact prediction was defined as having precisely the same component size [5]. The study received ethical approval from the Ethical Committee of Saarland, number 123/24.

### Statistical Analysis

Statistical analysis was performed using Python version 3.11.4 with the following packages: matplotlib, pandas, scipy, seaborn, and numpy in Jupyter Notebook version 6.5.4 environment. To allow for statistical analysis, the tibial component sizes were translated into a numerical scale starting at 1 and increasing in increments of 1. Femur components retained their numerical scale. The demographic description of the patient population was collated into a table. The duration of surgery and length of stay were included in the analysis. For comparing planned and implanted components, the normality of distribution was excluded and a Wilcoxon test was performed.

For the axis measurement comparison, the paired differences between consultant and 2D to 3D X-Atlas^®^ measurements were calculated. If there was a coronal plane (varus–valgus) discrepancy between those two measurements, first both planning documents were checked to exclude error during data collection, and then, the absolute difference was calculated (i.e., 2 degrees to varus vs. 3 degrees to valgus would yield a 5-degree difference). Senior consultant measurement were taken as the gold standard [6], while deviations were recorded as the 2D to 3D X-Atlas^®^ planning results. The distribution of the differences was plotted and 4 patients were marked as significant outliers. The measurement of the axis was re-checked by a senior consultant, and confirmed to be the same procedure measured previously. These were excluded from calculating the minimal significant bias of the study, but were included in the rest of the analysis. The normality of the distribution was confirmed by the Shapiro–Wilk test and visual plot. The results were plotted using the Bland–Altman method and a paired *t*-student test was run, with the mean difference and coronal plane deviation recorded. The threshold for statistical significance for all tests was established to be *p* = 0.05.

## 3. Results

### 3.1. Baseline Characteristics

A total of 55 patients were included in the study, 30 underwent right and 25 left total knee arthroplasty. Patients who underwent bilateral TKR, whose blinding was lost or received no preoperative planning, were not included. The baseline characteristics along with operation duration and length of stay for the investigated population can be found in Table 1.

### 3.2. Femoral Component Matching

Femur components were exactly matched by the planning provided by 2D to 3D X-Atlas^®^ in 46 (83.64%) cases. The prediction was accurate in a further eight (14.55%) cases and inaccurate only in one (1.82%). The femur component distribution is depicted by Figure 1, where the predicted and actual sizes are plotted over each other.

The Wilcoxon test statistic for the femoral component size comparison was found to be 18.00 with *p* = 0.5637 (>0.05), meaning that the difference between the predicted and implanted groups was not statistically significant.

### 3.3. Tibial Component Sizes

Tibial components were exactly matched by the planning for 39 (70.91%) patients, accurately for 12 (21.82%), and inaccurately for 4 (7.27%). The Wilcoxon test was run for the numerical translation of tibial component sizes and yielded a statistic equivalent to 13.00 with *p* = 0.0028 (<0.05), meaning that the difference between the predicted and implanted groups was statistically significant. The distribution of projected and implanted component sizes is portrayed by Figure 2.

### 3.4. Coronal Axis Analysis

A total of 51 patients were included in the analysis of bias and discrepancy between the 2D to 3D X-Atlas^®^ and the senior consultant plane measurements. Clinician measurements were used as the reference point, with positive deviation towards valgus and negative deviation towards varus. 2D to 3D X-Atlas^®^ planning has a minimal statistically significant bias of 0.83 degrees into varus. This is proven by the paired Student *t*-test carried out on the differences between robotic measurements and the clinician. It yielded a *p* = 0.0001. The Bland–Altman test for this result is shown in Figure 3. The inclusion of the four censored points would increase the bias towards varus, showing a potentially bimodal distribution in the dataset. Moreover, the deviation of all those measurements from the gold standard was over 4 degrees, making their axis assessment highly unreliable.

The mean final alignment of the knee in the coronal axis was 0.37 degrees in varus, with a standard deviation of 2.49.

## 4. Discussion

This study evaluated the accuracy of the ROSA^®^ Knee System and 2D to 3D X-Atlas^®^ (Zimmer Biomet, Warsaw, IN, USA) in predicting the sizes of the femoral and tibial components and measuring the knee axis in primary total knee replacement (TKR) compared to the measurements performed by senior surgeons. The literature appears divided on the improvements of mechanical alignment associated with robotic-assisted TKA. A meta-analysis of nine randomised control trials showed that robotic-assisted TKA is associated with improved mechanical alignment, femoral coronal and sagittal and tibial sagittal outliers compared to conventional RTK [7]. In contrast, another meta-analysis found no difference in the postoperative outcomes of the femoro–tibial angle between robotically assisted RTK and conventional methods [8]. Within different robotic-assisted systems, ROSA has been shown to have comparable component positioning accuracy to the CT-based, saw cutting robotic system (MAKO) [4]. The comparison between different robotic systems can, however, point to safer and more optimal solutions. This study further expands our understanding of the technology, specifically assessing femoral and tibial component sizing accuracy by ROSA^®^, as well as the built-in bias in the axis alignment of the robot. We report on the baseline characteristics of the population operated on in Table 1. The average post-RA-TKR stay (6.9 days) was slightly shorter than the overall average of 7.1 days for any other arthroplasty at the center in that time period. It does however lie within one standard deviation from the mean.

### 4.1. Accuracy of Femoral and Tibial Component Size Prediction

The ROSA Knee System demonstrated a high level of accuracy in predicting femoral component sizes, with an exact match in 83.64% of cases and an accurate prediction (within one size) in 98.18%. The statistically insignificant difference between predicted and implanted femoral components underscores the reliability of ROSA^®^ in femoral component sizing. Our results have found ROSA^®^ planning to be more accurate compared to another study reporting on femoral component sizing, which was accurate in 87% of cases [5]. This precision is essential, as improperly sizing the femoral component can lead to suboptimal implant function, increased wear, and complications such as early implant failure or the need for revision surgery [9]. The statistically insignificant discrepancy between predicted and fitted femoral components suggests that the 2D to 3D X-Atlas^®^ can significantly enhance preoperative planning and surgical accuracy in this regard.

In contrast, the accuracy of the tibial component matching was lower, with exact matches in 70.91% of cases and an accurate prediction in 92.73%. It is important to note that the difference between predicted and implanted tibial sizes was statistically significant. Along the lower precision compared to the femoral component sizing, this highlights that surgeons should exercise caution when relying on X-Atlas^®^ planning. Our results also show that the 2D to 3D X-Atlas^®^ was more accurate for tibial component prediction compared to previous groups which reported an accuracy of 77% compared to our 92.73% [5]. Those studies also report a lower accuracy of the tibial component (77%) than the femoral component matching (87%) [5]. The presence of some inaccurate predictions (7.27%) may warrant further investigation of the factors contributing to this variability, such as patient-specific anatomical differences, the positioning of optical guidance tools, or very significant axis deviation leading to inaccurate planning.

### 4.2. Bias in Axis Measurements

The pooled analysis from a recent systematic review showed that ROSA^®^-TKR component positioning had a 0.61–1.87° accuracy and a precision in the range of 0.97–1.34° for both coronal and sagittal parameters [3]. The prediction of ROSA^®^ software has also been shown to be poor, with low correlations between predicted and actual angles in the coronal plane and no correlation for angles in the sagittal plane [1]. Comparable findings were observed in other robotic-assisted TKAs such as the CT-guided MAKO system [4]. Our study further expands on coronal accuracy and demonstrates a minimal statistically significant bias of 0.83° towards varus in the 2D to 3D X-Atlas^®^ axis measurements compared to those of an experienced senior consultant (*p* = 0.0001). Despite this varus bias in preoperative measurement, our results show that the ROSA^®^-TKR results in deviations within the ±3° of neutral alignment are generally deemed acceptable in TKR [10]. In fact, our final postoperative coronal axis measurements had a mean of 0.37° (SD = 2.48) into varus. If this varus alignment is significant, it can be associated with an increase in medial compartment loading, which may predispose patients to accelerated wear and implant failure over time.

We have also demonstrated a potential bimodal distribution in the varus deviation, with four points that had significantly more bias than the normally distributed data. These suggest a potential outlier group that receives an unreliable axis measurement. Due to the small size of that group, the factors leading to this unreliability could not be identified. It should be noted that, for around 7% of the cases, the axis measurement by the 2D to 3D X-Atlas^®^ can be erroneous (more than 4° measurement difference).

### 4.3. Implications for Surgical Outcomes and Future Directions

The high accuracy of femoral component sizing and the acceptable performance in tibial component prediction suggest that robotic-assisted surgery with the ROSA^®^ system along with the 2D to 3D X-Atlas^®^ used can improve the precision of TKR procedures. While small, the varus bias observed in axis measurements highlights an area for future investigation, as its significance is unknown. One suggested implication is that it may result in a valgus postoperative knee axis if the 2D to 3D X-Atlas^®^ planning is used alone for preoperative planning.

In future studies, larger sample sizes and longer follow-up periods are needed to assess the clinical significance of the ROSA^®^ system’s bias and its impact on patient-reported outcomes, implant survival, and complication rates. Furthermore, comparing the ROSA^®^ system with other robotic-assisted technologies could provide insights into the relative advantages and limitations of different systems in TKR. Surgeons should remain vigilant about potential biases in robotic measurements and continue to use their clinical expertise in conjunction with technology to optimize patient outcomes.

## 5. Conclusions

The 2D to 3D X-Atlas^®^ shows strong potential in accurately predicting component sizes in TKR, particularly for femoral components. However, its bias towards varus in the axis measurements, whilst small, is statistically significant and should be noted when using data obtained for future operation planning. Despite this bias, our results show that the final coronal-plane measurement is within the accepted standard of ±3° degrees. Continuous refinement and further research on its clinical implications will be critical for optimising RA-TKR.

## Figures and Tables

**Figure 1 jcm-14-01698-f001:**
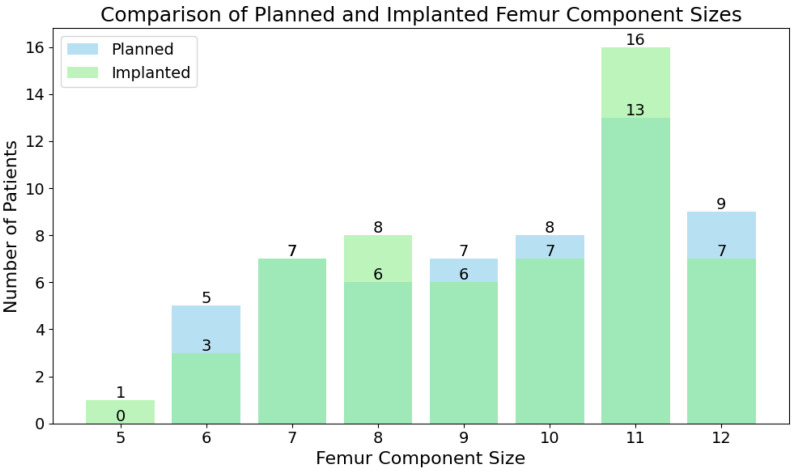
Predicted and implanted femoral component size distribution.

**Figure 2 jcm-14-01698-f002:**
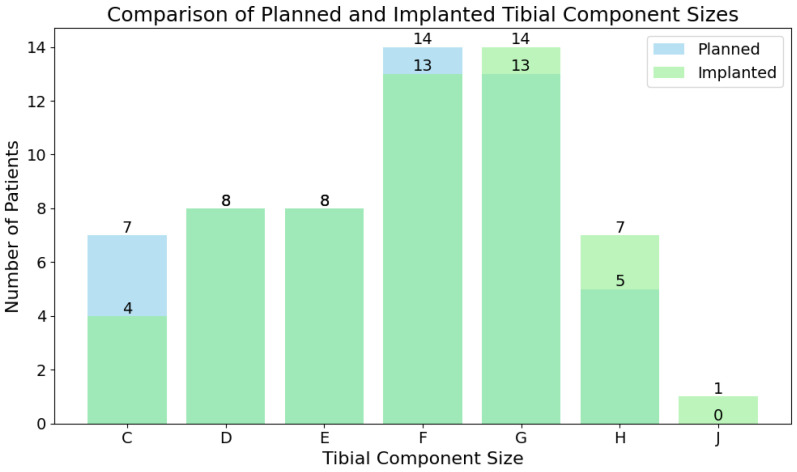
Predicted and implanted tibial component size distribution.

**Figure 3 jcm-14-01698-f003:**
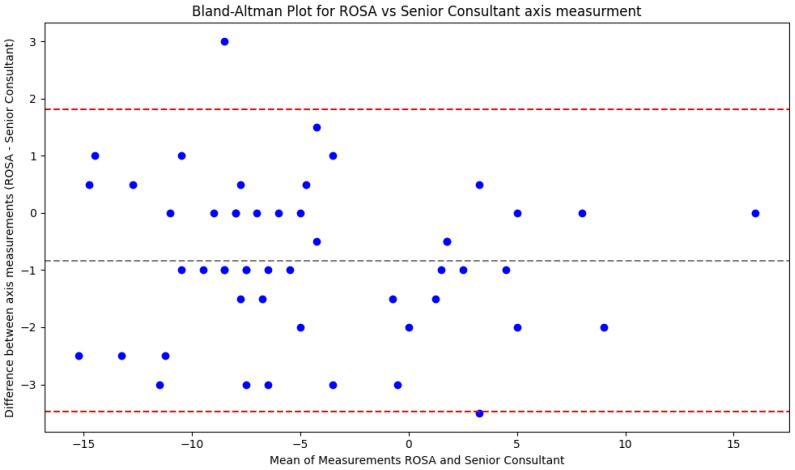
Bland–Altman graph for the distribution of means vs. the differences between the axes measured by the 2D to 3D X-Atlas and a senior consultant. Gray line represents the mean difference (if negative, pointing in the varus direction), while red lines represent one standard deviation. The graph is considered valid if most points lie within one standard deviation of the mean.

**Table 1 jcm-14-01698-t001:** Patient demographics and procedure details.

Parameter	Mean (SD)
Age in years (SD)	70.6 (9.0)
Female	32.7% (18) ^1^
BMI (SD)	29.4 (4.4)
ASA score (SD)	2.3 (0.6)
Operation time in minutes (SD)	111.9 (20.1)
Length of stay in days (SD)	6.9 (2.2)
Varus limb deformity	69% (38) ^1^
Valgus limb deformity	29% (16) ^1^
No limb deformity	2% (1) ^1^

^1^ Percentage (*n*).

## Data Availability

Data can be provided by the authors upon request.

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
