# Peer review of "Inherent Bias in ROSA® Zimmer Biomet Pre-Op Planning Using 2D to 3D X-Atlas® Coronal Knee Axis Measurement"

_jcm, 2025, doi:10.3390/jcm14051698_

Round 1

Reviewer 1 Report

Comments and Suggestions for Authors

The authors examine the accuracy of the X-atlas in determining final implant size in total knee arthroplasty using the Rosa knee system in addition to comparing lower limb axis with measurements derived from senior consultant surgeons. They demonstrate good prediction of implant size overall, however with a bias towards varus final lower limb alignment.

I feel the manuscript is well written and overall adds some value to the existing literature base. I would recommend minor revision with comments as below.

Abstract – Sufficiently informative, satisfactory length

Introduction – Line 24, edit “This method comparable or better outcomes than conventional TKR” to “This method achieves comparable or better outcomes than conventional TKR”. Otherwise good / concise.

Methods – Generally with lower limb alignment studies whereby an operator undertakes measurement, I would expect a degree of validation. A common approach would be where two operators would undertake the measurements separately and correlation verified by a tool such as Pearson’s correlation and the mean of the two figures used for the final analysis. It is not clear that any verification took place in this study. I would advise the authors to clarify this.

Statistical analysis – Could the authors clarify why “senior consultant” measurements are taken as the gold standard whereby the X-Atlas tool does reference the whole lower limb (ie. hip – knee – ankle).

Results – Presented well overall, clear graphs and tables

Author Response

Comment 1: Introduction – Line 24, edit “This method comparable or better outcomes than conventional TKR” to “This method achieves comparable or better outcomes than conventional TKR”. Otherwise good / concise.

Response 1: Thank you for pointing out this mistake, we must have omitted that during final revision and editing of the manuscript. Following changes have been made to the manuscript: “Some studies suggest that this method has at least comparable outcomes to conventional TKR [1][2 ].”

Comment 2: Methods – Generally with lower limb alignment studies whereby an operator undertakes measurement, I would expect a degree of validation. A common approach would be where two operators would undertake the measurements separately and correlation verified by a tool such as Pearson’s correlation and the mean of the two figures used for the final analysis. It is not clear that any verification took place in this study. I would advise the authors to clarify this.

Response 2: This is a valid point, our understanding of the Reviewers comment is that the preoperative measurement would require validation and measurement by two senior consultants before being taken as gold-standard. We have used the Sectra 2D Planning System tool for taking the measurement, and the consultant operating was taking the measurement pre-operatively, unfortunately no second validation by a different consultant was performed. If the Reviewer question refers to postoperative alignment, this was taken using the ROSA system.

Comment 3: 
Statistical analysis – Could the authors clarify why “senior consultant” measurements are taken as the gold standard whereby the X-Atlas tool does reference the whole lower limb (ie. hip – knee – ankle).

Response 3: Thank you for asking this important question - X-Atlas tool is indeed taking the whole lower limb into account: hip-knee-ankle, but does use soft-tissue mounted guides for reference, and its method nor code of determining the axis is not open-source, so we can not be certain how it establishes the axis measurement. We know however, that we can use the current gold standard - whole leg-length X-ray measurement of the mechanical axis, which was accurately performed by our most experienced members of staff - “senior consultant”. Following changes to manuscript have been made: Materials and Methods: “The same images used for 2D to 3D X-Atlas® planning were used to perform knee axis measurement by a senior consultant, which is the current gold standard [5]. “

Reviewer 2 Report

Comments and Suggestions for Authors

Thank you for the opportunity to review the manuscript: “Inherent bias in coronal knee axis measurement by ROSA Zimmer Biomet pre-op planning”. This study raises a current issue concerning the accuracy of predicting implant sizing and axis measurements in coronal view. The authors analyzed 55 patients who underwent ROSA – TKAs and compared pre-procedural and postoperative measurements using X-Atlas 2D to 3D to those performed by a senior consultant. I see great potential in this study, however, it requires major revisions to improve quality before publication.

General comments:

This study consisted of all (as the authors wrote in the material and methods section) ROSA TKAs performed over 2 years. As the authors wrote surgeries were done by 4 high-volume surgeons (>100 TJAs / year). Only 55 TKA were included. If we calculate a minimum of 800 TJAs for these 4 surgeons in a 2-year period (4 surgeons x 100 TJAs x2 years), ROSA-TKA was performed only in a maximum of 6.75% of cases. It is less than 7 ROSA TKA per year for one surgeon. It is not much.

Introduction:

General comment: The introduction is very short.

  • “This method comparable or better outcomes than conventional TKR” – why better? The references that you attached indicate a comparison. Please explain.
  • The authors included some information about shorter hospital stays and better 6-month post-operative outcomes for pain and disability. The subject of this study is related to radiological accuracy. I suggest giving a short comment about the preoperative accuracy of digital/analog templating and the general accuracy of robotics’ TKA with different systems.

The aim of this study should be stated clearly. In the introduction section was noted that : “In our study we check for predicted versus implanted component size matching and look for biases (..) by the 2D to 3D X-Atlas® planning” and in the methods section was mentioned that : “Images used for 2D to 3D X-Atlas® planning were used to perform knee axis measurement by a senior consultant” This was then compared to values obtained from X-Atlas®”. I suggest defining the aims of this study clearly at the end of the introduction section. I understand you made 2 comparisons: 1) X-Atlas vs senior consultant (preoperatively); 2) X-Atlas implant size predictions vs postoperative final sizes. I’m right?

Materials and methods:

  • This study has prospective or retrospective character? Please specify in this section.
  • I don’t understand why you excluded 4 patients with “significant outliers”. The group is small, and you still decided to exclude 7% of patients. This can cause wrong assessment and significant bias in the analysis. It can also question and overstate the accuracy of X-atlas measurements. It requires an explanation in the manuscript and if you did not analyze the results obtained in these 4 patients you should mention these patients met the “exclusion criteria”.
  • There is a need to add some information about the patients’ clinics – a type of deformation (varus/valgus) and grade of OA e.g. Kellgren-Lawrence scale. It can give information this group of patients was homogenous.
  • “Duration of surgery and length of stay were included in the analysis.” – for one group you just presented the received data without any comparison / analysis. There is no comment about these data in the manuscript.
  • “For comparing planned and implanted elements a one-way ANOVA test was performed” – I’m not sure that ANOVA (test no information about parametric / non-parametric) is appropriate for that comparison if we have only two lists of data.
  • Why did you use two tests for testing the normality of distribution? The Shapiro-Wilk test is preferred for small samples (n is less than or equal to 50). For larger samples, the Kolomogrov-Smirnov test is recommended. Please explain.

Results:

  • As I mentioned before I’m not sure the ANOVA test is appropriate for the comparison of two groups. There is no information is it a parametric or non-parametric test. For comparison of the two groups, we have different statistical tests depending on the data present normal (or not) distribution.
  • You compared planned coronal alignment X-atlas vs senior consultant, and these results were statistically different (p<0.0001). Did you perform the analysis of which one (The x-atlas or consultant) was closer to the final postoperative alignment?

Discussion:

  • “The ROSA Knee System demonstrated a high level of accuracy in predicting femoral component sizes, with an exact match in 83.64% (…)” and then: “In contrast, the tibial component accuracy was slightly lower, with exact matches in 70.91% of cases” – I can’t completely agree this accuracy is a “high level” and >12% of the difference between femur and tibia is slightly lower. You researched the expensive equipment that increases the total costs of the TKA procedure, which should present overall accuracy higher even if only X-rays (not CT) were done. There are a lot of papers investigating the accuracy of digital templating in TKA, that you should compare with your results in the discussion section. Moreover, you focused on the ROSA system. Several papers investigated the accuracy of preoperative templating in different robotic systems that can be discussed concerning ROSA.
  • Chapter 4.3 repeats again the obtained results of implant sizing and coronal alignment (as in the results section, and 4.2 chapter) and has the nature of the authors' reflections without referencing any studies concerning implications for surgical outcomes. I counted that the authors wrote 5 times in the text that ROSA may lead to "varus bias observed in axis measurements", is it necessary?

References:

  • Only 7 references? I recommend citing more studies about the accuracy of robotics in TKA. E.g “Robotics in TKA” has 815 results in PubMed, you have a lot of literature on this subject.

Comments on the Quality of English Language

Can be improved by a native speaker.

Author Response

Comment 1: This study consisted of all (as the authors wrote in the material and methods section) ROSA TKAs performed over 2 years. As the authors wrote surgeries were done by 4 high-volume surgeons (>100 TJAs / year). Only 55 TKA were included. If we calculate a minimum of 800 TJAs for these 4 surgeons in a 2-year period (4 surgeons x 100 TJAs x2 years), ROSA-TKA was performed only in a maximum of 6.75% of cases. It is less than 7 ROSA TKA per year for one surgeon. It is not much.

Response 1: Thank you for pointing out this important fact. We included patients who received X-Atlas Pre-Operative planning, which were unfortunately not all the patients who underwent ROSA-TKR. X-Atlas planning required additional lateral whole-leg-length X-ray that was not routinely performed by the radiology department at the clinic. The initial cases which were performed without the X-Atlas preoperative planning were also not included as those did not match the inclusion criteria. The following changes have been made to better outline the actual scope of the test group: Material and Methods: “All patients in a single center who received the X-Atlas® preoperative planning and underwent robotic-assisted primary total knee replacement between February 2022 and May 2024 using the ROSA® Knee System (Zimmer Biomet, Warsaw, IN) were included.”

Comment 2: General comment: The introduction is very short.

Response 2: We have expanded on the issues raised by the Reviewer. These parts are marked in answer to comments below.

Comment 3: “This method comparable or better outcomes than conventional TKR” – why better? The references that you attached indicate a comparison. Please explain.

Response 3: Following changes have been made to the manuscript: “Robotic surgical assisted technology (ROSA®) is well-established for total knee replacements (TKR). Some studies suggest that this method has at least comparable outcomes to conventional TKR [1][2 ]. Further analysis of 17 studies has shown that the ROSA system results in improved clinical outcomes within 1 year of surgery [3 ].”

Comment 4: The authors included some information about shorter hospital stays and better 6-month post-operative outcomes for pain and disability. The subject of this study is related to radiological accuracy. I suggest giving a short comment about the preoperative accuracy of digital/analog templating and the general accuracy of robotics’ TKA with different systems.

Response 4: The following has been added: “ROSA®-TKR has been shown to predict both femoral and tibial resections more accurately than conventional methods [3 ]. This enhanced accuracy is comparable to that of other robotic-assisted technologies, such as the MAKO robotic-arm assisted technology, despite using two X-ray projections rather than a CT-scan for planning [ 4 ]. Coronal tibial and femoral cuts are predicted more accurately with ROSA® than sagittal ones [1][5].”

Comment 5: The aim of this study should be stated clearly. In the introduction section was noted that : “In our study we check for predicted versus implanted component size matching and look for biases (..) by the 2D to 3D X-Atlas® planning” and in the methods section was mentioned that : “Images used for 2D to 3D X-Atlas® planning were used to perform knee axis measurement by a senior consultant” This was then compared to values obtained from X-Atlas®”. I suggest defining the aims of this study clearly at the end of the introduction section. I understand you made 2 comparisons: 1) X-Atlas vs senior consultant (preoperatively); 2) X-Atlas implant size predictions vs postoperative final sizes. I’m right?

Response 5: This is a very important point and we have addressed it accordingly - the aims of the study as the Reviewer has kindly pointed out were to: 1. Compare the measurements performed by X-Atlas which is essentially a black-box (not open-source, and not disclosing its method of measurement) to the current gold standard for mechanical knee axis alignment measurement which is whole-length X-ray. Luckily the same picture can be used for both measurements (X-Atlas) and senior consultant, preventing any potential bias or mistakes due to rotation, or picture-dependent changes. 2. Final size prediction accuracy. Following changes have been made to address these concerns:

The following sentence was changed from: “In our study we check for predicted versus implanted component size matching and look for biases in coronal knee axis measurement by the 2D to 3D X-Atlas® planning.”

To:“The aims of the study were:

  1. To measure and quantify the accuracy of the pre-operative knee axis alignment measurement taken by the 2D to 3D X-Atlas® planning compared to the current gold-standard of whole-leg-length X-ray measurement performed by a senior consultant [5].
  2. To assess the accuracy of component size prediction compared to used implants by the 2D to 3D X-Atlas® pre-operative planning.”

Comment 6: This study has prospective or retrospective character? Please specify in this section.

Response 6: Following phrase was added to the manuscript Materials and Methods: “Patients were included retrospectively based on the availability of the pre-operative X-Atlas® planning.”

Comment 7: I don’t understand why you excluded 4 patients with “significant outliers”. The group is small, and you still decided to exclude 7% of patients. This can cause wrong assessment and significant bias in the analysis. It can also question and overstate the accuracy of X-atlas measurements. It requires an explanation in the manuscript and if you did not analyze the results obtained in these 4 patients you should mention these patients met the “exclusion criteria”.

Response 7: Given sample size and the risk of under/overestimation of the bias, I agree with the reviewer that the need of censoring 4 points is a pity. Having defined the scope more precisely this approach can be better explained. We wanted to determine the existence of, and if possible, quantify the bias within the X-Atlas method. In order to achieve that in a measurable and reliable way, we can identify the “minimal true bias” based on the normally distributed results. It is an established method in statistics, to censor significant outliers in otherwise normally distributed data if there is a valid explanation for that action. This can be presented graphically using two plots, which I attach to prove that the data was normally distributed apart from 4 significant outliers in Varus. Now, this can mean that there is a bimodal distribution, or those are exceptions in the overall population. It would be interesting to look for that distribution in a bigger sample. The X-rays for those 4 data points were reviewed and the measurement was repeated yielding the same result by a different senior consultant. We decided to treat them as significant outliers in the context of bias calculation, and analyse the otherwise normally distributed data. We should have commented on the 4 points with massive difference between the X-Atlas and senior consultant measurment, and pointing to a potential bimodal distribution of bias, however this could not be statistically analysed due to small size of the erroneous sample. (App. 1, App. 2) We can draw conclusions of a minimal statistically significant bias of 0.83 degrees into Varus. I agree with the rfeviewer that this can be an underestimation of the measurement error, however we could not give a reliable statistically significant number without a much larger sample to account for those 4 points.

It is a normal practice in the Bland-Altmann method to censor significant outliers (points far beyond the red dashed lines). As this is a graphical method of showing statistical significance of data trends, it is less accurate than the paired t-test used to analyse the sample. For reference, I attach the uncensored Bland-Altmann (App. 3).

To address this problem the following modifications to the Manuscript have been made:
Materials and methods: Statistical analysis: “The distribution of the differences was plotted and 4 patients were marked as significant outliers. Measurement of the axis was re-checked by a senior consultant, and confirmed to be the same as measured previously. These were excluded from calculating the minimal significant bias of the study, but included in the rest of the analysis. The normality of the distribution was confirmed by the Shapiro-Wilk test and visual plot. The results were plotted using the Bland-Altman method and a paired t-student test was run, with mean difference and coronal plane deviation recorded. The threshold for statistical significance for all tests was established at p = 0.05.”
Results: Coronal axis analysis: “2D to 3D X-Atlas® planning has a minimal statistically significant bias of 0.83 degrees into varus. This is proven by the paired t-student test on the differences between robotic measurement and the clinician. It yielded a p = 0.0001. Bland-Altman test for this result is shown in Figure 3. Inclusion of the 4 censored points would increase the bias towards varus, they showed a potentially bimodal distribution in the dataset. Moreover, the deviation of all those measurements from the gold-standard was over 4 degrees, making their axis assessment highly unreliable. ”
Discussion: “We have also demonstrated a potential bimodal distribution in the varus deviation, with 4 points that had significantly more bias than the normally distributed data. They suggest a potential outlier group that receives an unreliable axis measurement. Due to the small size of that group, the factors leading to this unreliability could not be identified. It should be noted that for around 7% of the cases the axis measurement by 2D to 3D X-Atlas® can be erroneous (more than 4 degrees measurement difference). “

Comment 8: There is a need to add some information about the patients’ clinics – a type of deformation (varus/valgus) and grade of OA e.g. Kellgren-Lawrence scale. It can give information that this group of patients was homogenous.

Response 8: 
A valid point has been made in terms of varus-valgus deformity, we have added the following to patient population description table:
“Varus limb deformity 69% (38)1

Valgus limb deformity 29% (16)1

No limb deformity 2% (1)1”

Kellegren-Lawrence scale is not normally commented on in the centre, and it has no implication on the precision of planning, so this data was not readily available. If required we can obtain that data.

Comment 9: “Duration of surgery and length of stay were included in the analysis.” – for one group you just presented the received data without any comparison / analysis. There is no comment about these data in the manuscript.

Response 9: 
Indeed, these are placed as baseline characteristics to establish how our patient group and standards of care looked like. 

Discussion: “The average post-RA-TKR stay (6.9 days) was slightly shorter than the overall average of 7.1 days for any other arthroplasty at the centre in that time period. It does however lie within the one standard deviation from the mean.”

Comment 10: “For comparing planned and implanted elements a one-way ANOVA test was performed” – I’m not sure that ANOVA (test no information about parametric / non-parametric) is appropriate for that comparison if we have only two lists of data.

Response 10: The reviewer raises an important issue about ANOVA being used. I see that the reviewer is right in their doubt for this method. To combat that, we have checked the normality of distribution of all 4 compared groups, and indeed, the data is not normally distributed, which is proven by Shapiro-Wilk. The following changes were made in text: “For comparing planned and implanted elements normality of distribution was excluded and a Wilcoxon test was performed.”

Comment 11: Why did you use two tests for testing the normality of distribution? The Shapiro-Wilk test is preferred for small samples (n is less than or equal to 50). For larger samples, the Kolomogrov-Smirnov test is recommended. Please explain.

Response 11: That is indeed true, but Shapiro-Wilk is considered reliable up to n=2000, therefore as our sample lies much closer to 50 we preferred the Shapiro-Wilk test. For completeness the Kologomorov-Smirnov test was performed, and graphic QQ plot as well as plotting of distribution are attached in App. 1.

Comment 12: As I mentioned before I’m not sure the ANOVA test is appropriate for the comparison of two groups. There is no information is it a parametric or non-parametric test. For comparison of the two groups, we have different statistical tests depending on the data present normal (or not) distribution.

Response 12: Thanks to the reviewer comments we were able to identify the correct test to use to compare our data - a paired sample test for non-normally distributed data that checks whether the two samples came from the same dataset distributions. It did identify an error in our analysis. There is in fact a statistically significant difference in predicting the final Tibial component size by the X-Atlas planning. This impacts overall outcome of the study and the manuscript has been changed accordingly:

Deleted: “ANOVA test for difference between planned and actual sizes of the femoral component yielded a p = 0.96, making difference between those two groups statistically insignificant.”

Replaced by: “Wilcoxon test statistic for Femur component size comparison = 18.00 with p = 0.5637 (>0.05) meaning that the difference between the predicted and implanted groups was not statistically significant.”

Deleted: “One-way ANOVA comparing the predicted and implanted tibial sizes translated into numerical values produced a p = 0.27.”

Replaced by: “Wilcoxon test run for numerical translation of Tibial component sizes yielded a statistic of = 13.00 with p = 0.0028 (<0.05) meaning that the difference between the predicted and implanted groups was statistically significant.”

Comment 13: You compared planned coronal alignment X-atlas vs senior consultant, and these results were statistically different (p<0.0001). Did you perform the analysis of which one (The x-atlas or consultant) was closer to the final postoperative alignment?

Response 13: It is an important question, however due to an overall aim of reducing the final postoperative angle to 0, there are too many factors at play to draw significant conclusions in such a small sample, i.e. intraoperative modification of the component placement and balancing of relevant soft tissue.

Comment 14: “The ROSA Knee System demonstrated a high level of accuracy in predicting femoral component sizes, with an exact match in 83.64% (…)” and then: “In contrast, the tibial component accuracy was slightly lower, with exact matches in 70.91% of cases” – I can’t completely agree this accuracy is a “high level” and >12% of the difference between femur and tibia is slightly lower. You researched the expensive equipment that increases the total costs of the TKA procedure, which should present overall accuracy higher even if only X-rays (not CT) were done.

Response 14: Following changes have been made to the manuscript: Accuracy of Tibial and Femoral Component matching: “In contrast, the accuracy of the tibial component matching was lower, with exact matches in 70.91% of cases and an accurate prediction in 92.73%. It is important to note, that the difference between predicted and implanted tibial sizes was statistically significant. Along the lower precision compared to the femoral component sizing, this highlights an important caution for surgeons relying on X-Atlas® planning. Our results also show that 2D to 3D X-Atlas® was more accurate for tibial component prediction compared to previous groups which reported an accuracy of 77% compared to our 92.73% [5 ].”

Comment 15: There are a lot of papers investigating the accuracy of digital templating in TKA, that you should compare with your results in the discussion section. Moreover, you focused on the ROSA system. Several papers investigated the accuracy of preoperative templating in different robotic systems that can be discussed concerning ROSA.

Response 15: Following changes have been made to the manuscript: "The literature appears divided on the improvements of mechanical alignment associated with robotic-assisted TKA. A meta-analysis of nine randomised control trials showed that robotic-assisted TKA is associated with improved mechanical alignment, femoral coronal and sagittal and tibial sagittal outliers compared to conventional RTK [7]. In contrast, another meta-analysis found no difference in the postoperative outcomes of the femoro-tibial angle between robotically assisted RTK and conventional methods [ 8]. Within different robotic-assisted systems, ROSA has been shown to have equal component positioning accuracy, as the CT-based, saw  cutting robotic system (MAKO) [ 4]. The comparison between different robotic systems can on the other hand point to more optimal and safer solutions. This study further expands the understanding of the technology, specifically assessing femoral and tibial component sizing accuracy by ROSA® as well as the built-in bias in axis alignment of the robot.”

Comment 16: Chapter 4.3 repeats again the obtained results of implant sizing and coronal alignment (as in the results section, and 4.2 chapter) and has the nature of the authors' reflections without referencing any studies concerning implications for surgical outcomes. I counted that the authors wrote 5 times in the text that ROSA may lead to "varus bias observed in axis measurements", is it necessary?

Response 16: The section was changed accordingly: “The high accuracy of femoral component sizing and the acceptable performance in tibial component prediction suggest that robotic-assisted surgery with the ROSA® system along with 2D to 3D X-Atlas® use can improve the precision of TKR procedures. While small, the varus bias observed in axis measurements highlights an area for future investigation as its significance is unknown. One suggested implication is that it may result in a valgus postoperative knee axis if the 2D to 3D X-Atlas® planning is used alone for preoperative planning. In future studies, larger sample sizes and longer follow-up periods are needed to assess the clinical significance of the ROSA® system’s bias and its impact on patient-reported outcomes, implant survival, and complication rates. Furthermore, comparing the ROSA®system with other robotic-assisted technologies could provide insight into the relative advantages and limitations of different systems in TKR. Surgeons should remain vigilant about potential biases in robotic measurements and continue to use their clinical expertise in conjunction with technology to optimize patient outcomes.”

Comment 17: Only 7 references? I recommend citing more studies about the accuracy of robotics in TKA. E.g “Robotics in TKA” has 815 results in PubMed, you have a lot of literature on this subject.

Response 17: 3 more references have been added to adjust to the length and literature breadth that the paper covers.
"4. Zhou, G.; Wang, X.; Geng, X.; Li, Z.; Tian, H. Comparison of alignment accuracy and clinical outcomes between a CT-based, saw cutting robotic system and a CT-free, Jig-guided robotic system for total knee arthroplasty. Orthopaedic surgery 2024. "

"7. Daoub, A.; Qayum, K.; Patel, R.; Selim, A.; Banerjee, R. Robotic assisted versus conventional total Knee Arthroplasty: A systematic review and meta-analysis of Randomised Controlled Trials - Journal of Robotic surgery. SpringerLink 2024. "

"Version February 19, 2025 submitted to J. Clin. Med. 8 of 88. Kaneko, T.; Igarashi, T.; Takada, K.; Yoshizawa, S.; Ikegami, H.; Musha, Y. Robotic-assisted total knee arthroplasty improves the outlier of rotational alignment of the tibial prosthesis using 3DCT measurements. The Knee 2021, 31, 64–76. https://doi.org/https://doi.org/10.1016/j.knee.2021.05.009."

Round 2

Reviewer 2 Report

Comments and Suggestions for Authors

Thank you for the opportunity to review the corrected version of the manuscript. 

The authors answered all my questions and made appropriate changes to the text according to my suggestions.

2 small suggestions:

- remove "COVID-19" from keywords

-change "no limb deformity" to "neutral limb axis" in Table 1.

I have no other concerns. Good job.

Comments on the Quality of English Language

Quality was improved.